# Developing a Vaccine to Block West Nile Virus Transmission: In Silico Studies, Molecular Characterization, Expression, and Blocking Activity of *Culex pipiens* mosGCTL-1

**DOI:** 10.3390/pathogens10020218

**Published:** 2021-02-17

**Authors:** Hasan Bakhshi, Mehdi Fazlalipour, Javad Dadgar-Pakdel, Sedigheh Zakeri, Abbasali Raz, Anna-Bella Failloux, Navid Dinparast Djadid

**Affiliations:** 1Malaria and Vector Research Group (MVRG), Biotechnology Research Center (BRC), Pasteur Institute of Iran, Pasteur Ave., Tehran 1316943551, Iran; bakhshi@pasteur.ac.ir (H.B.); dadgar21@yahoo.com (J.D.-P.); zakeris@yahoo.com (S.Z.); 2Department of Arboviruses and Viral Hemorrhagic Fevers (National Ref Lab), Pasteur Institute of Iran, Pasteur Ave., Tehran 1316943551, Iran; mfp.virology@gmail.com; 3Trauma Research Center, Sina Hospital, Tehran University of Medical Sciences, Hassan Abad Square, Imam Khomeini Avenue, Tehran 1136746911, Iran; 4Institut Pasteur, Department of Virology, Arboviruses and Insect Vectors, 25 rue Dr. Roux, CEDEX 15, 75724 Paris, France

**Keywords:** *Culex pipiens*, *mosGCTL-1*, transmission blocking vaccine, West Nile virus

## Abstract

Background: Mosquito galactose-specific C-type lectins (mosGCTLs), such as mosGCTL-1, act as ligands to facilitate the invasion of flaviviruses like West Nile virus (WNV). WNV interacts with the *mosGCTL-1* of *Aedes aegypti* (Culicidae) and facilitates the invasion of this virus. Nevertheless, there is no data about the role of mosGCTL-1 as a transmission-blocking vaccine candidate in *Culex pipiens*, the most abundant Culicinae mosquito in temperate regions. Methods: Adult female *Cx. pipiens* mosquitoes were experimentally infected with a WNV infectious blood meal, and the effect of rabbit anti-rmosGCTL-1 antibodies on virus replication was evaluated. Additionally, in silico studies such as the prediction of protein structure, homology modeling, and molecular interactions were carried out. Results: We showed a 30% blocking activity of *Cx. pipiens mosGCTL-1* polyclonal antibodies (compared to the 10% in the control group) with a decrease in infection rates in mosquitoes at day 5 post-infection, suggesting that there may be other proteins in the midgut of *Cx. pipiens* that could act as cooperative-receptors for WNV. In addition, docking results revealed that WNV binds with high affinity, to the *Culex* mosquito lectin receptors. Conclusions: Our results do not support the idea that mosGCTL-1 of *Cx. pipiens* primarily interacts with WNV to promote viral infection, suggesting that other mosGCTLs may act as primary infection factors in *Cx. pipiens.*

## 1. Introduction

Arthropod-borne diseases contribute to the epidemics that disrupt health security [1], representing nearly 17% of all infectious diseases [2]. Mosquito-borne diseases such as malaria, filariasis, dengue, West Nile fever, chikungunya, yellow fever, Japanese encephalitis, and Zika fever are responsible for 700 million cases; and more than one million deaths each year [3]. For a long time, there was a wrong thought that mosquito-borne viruses like West Nile virus (WNV), Zika virus (ZIKV), dengue virus (DENV), and chikungunya virus (CHIKV) threaten only developing countries. However, these emerging and re-emerging viruses are spreading beyond their natural range of distribution, posing threats to temperate regions as well [4].

Only a few vaccines related to mosquito-borne diseases such as Dengvaxia against DENV and the 17D against yellow fever virus (YFV) are available [5,6]. Control of such diseases relies primarily on using insecticides [7]. Since the 1970s, this strategy is confronted with the emergence of insecticide-resistant populations within mosquitoes [8], stressing the need to develop new promising approaches like transmission-blocking vaccines (TBVs). TBVs against mosquito-borne diseases aim at preventing the transmission of arboviruses from infected to uninfected vertebrate hosts by targeting molecules expressed on the surface of pathogens or by targeting molecules expressed by mosquitoes. TBV candidates TBV candidates do not protect individuals against the infection, but they control the vector-borne diseases by blocking the transmission of pathogens among the population [9]. Despite the potential effectiveness of TBVs against viruses, this approach is still at its developmental stage and needs more improvements [4]. Mosquito-borne viruses such as WNV are enveloped with a cytoplasmic cycle for replication [10,11], adapted to both mosquitoes and vertebrates and thus, use evolutionarily conserved pathways shared by both hosts [12]. It has been revealed that some proteins in arthropods/mammals, which are named C-type lectins, are employed as receptors to facilitate the invasion of arboviruses [13].

In 2010, Cheng et al., reported that WNV induces the expression of mosquito galactose-specific C-type lectin-1 (*mosGCTL-1*), and interactions between WNV and *mosGCTL-1* are critical for infection of *Aedes aegypti* and *Culex quinquefasciatus*. In that study, generated antibodies against mosGCTL-1 of *Ae. aegypti* in rabbits inhibited WNV infection. Further, infection of *Cx. quinquefasciatus* (*CPIJ010995*) to WNV, resulted in up-regulation of mosGCTL-1 homolog in this mosquito [14]. 

WNV was isolated for the first time in 1937 from Uganda [15] and *Cx. pipiens* is considered as one of the primary enzootic vectors of WNV [16]. Currently, WNV is distributed in southern Asia, northern Australia, Africa, and sporadically in more temperate areas of Europe [17,18]. *Cx. pipiens* is widely distributed in many countries [19]. Regarding those are one of the medically important vectors and responsible for transmitting the WNV, characterizing and evaluating their *mosGCTL-1* as a TBV candidate is critical for understanding its role and interactions with the WNV and controlling the viral dissemination in the environment.

## 2. Results

### 2.1. Morphological and Molecular Identification of *Cx. pipiens*

Morphological identification and molecular analysis revealed that the collected *Cx. pipiens* mosquitoes from the Ghorogh forest belonged to form species *pipiens* (the sequence of Cytochrome oxidase subunit I (COI) gene of this strain was submitted to the GenBank under the KY646203 accession number). Nevertheless, *Cx. quinquefasciatus* was not identified in all the collected and examined *Culex* mosquito specimens.

### 2.2. Characterization of *Cx. pipiens mosGCTL-1*

Based on the conserved regions of the *mosGCTL-1* sequences of *Culex* species, a set of primers and an optimized amplification program were considered as described. The *mosGCTL-1* gene of *Cx. pipiens* encodes 159 amino acid residues; the length of intron for this gene in *Cx. pipiens* was 61 bp (MF361860 and MF361861 accession numbers), which sharing 93% of identity with the *mosGCTL-1* of *Cx. quinquefasciatus* (WGS: AAWU01017778) *mosGCTL-1* (Figure 1). 

### 2.3. Blocking Efficacy of *Cx. Pipiens rmosGCTL-1*

It was revealed that at 0, 2, 5, and 7 dpi number of the infected mosquitoes decreased at 5 and 7 dpi in the test group (14/20), (15/20) in comparison to the control group (18/20), (16/20) respectively (Figure 2; Appendix A).

### 2.4. Phylogenetic Prediction and Conserved Sequence

Clustal Omega alignment and subsequently phylogenetic tree analysis of the rmosGCTL-1 protein sequences of *Cx. pipiens* and other species indicated that this protein has very conserved residues, and it has predicted with more than 85% accuracy that *Culex* species have close root (Figure 3).

### 2.5. Prediction of the Pitope Avidity

Predicted interactions of the C-type lectin model (CTLM) epitopes with MHC-II using the NetMHCII server showed that the highest affinities among the evaluated CTLM peptides belonged to the HLA-DQA10102-DQB10602. Presentation capability by the common MHC-II alleles, which is determined by the affinity of the target peptides, is presented in Table 1. In this algorithm, peptide fragments, which are more compatible with specific MHC II alleles, form stronger binds with less energy. A smaller amount IC50 (nM) indicates a better presentation of peptide to the T helper cells. Therefore, better presentation of antigenic peptides by antigen presenting cells causes better activation of the immune system against West Nile virus.

### 2.6. Homology Modeling and Superimposition 

Based on the results of the SWISS-MODEL server, which was confirmed using the Phyre 2, C-type carbohydrate-recognition domain (CRD-4) from the macrophage mannose receptor; chain B (PDB accession No: 1EGI.B) had the highest structural identity (25.2%) to the target protein with a 97.5% confidence. Therefore this molecule was used as a reference molecule for beyond analysis. Superimposition of the reference molecule (1EGI) and CTLM showed that these two molecules were structurally 23.70% identical to each other, overall root-mean-square deviation (RMSD) was 0.567, and Q-score was 0.781. The primary structure of the model belonged to a family of calcium-dependent carbohydrate-binding proteins. The structure of this molecule consists of two α-helixes, five antiparallel ß-strands, and a wide flexible loop. Disulfide bonds due to the two cysteine residues stabilize the three-stranded antiparallel sheets which are conserved in the other C-type lectins such as lithostathine [20], tetranectin [21], factor IX/X-binding protein [22], CD94 [23], and Ly49a [24]. Unlike the central core of the protein, the “upper” segment, which contains an irregular and flexible loop (43° rotation concerning the core), has a significant difference from the other types of Mannose Binding Proteins (MBPs) and formed by residues ASP 84 to ASP 118 [25]. Some of the MBPs, unlike this molecule, have a compressed loop. Moreover, Ca^2+^ has special importance in the structure of MBPs and some of them have two calcium ions in their structure. One of them, which is deployed near the loop and named Ca^2+^ site 1, is auxiliary and induces the compacted loop structure [26]. However, Ca^2+^ site 2, which is called principal Ca^2+^, is very conserved among the all C-type lectin-like domains and plays an important role in binding to the sugar ligands. This calcium interacts directly with the β4 strand, the extended loop, and sugar ligands. Interestingly, the upper part of the loop of our target protein, which contains the residues from PHE 91 to ASP112, is involved in this interaction. The extended part of the loop which contains residues from ASP84 to THR 90 and GLU113 to ASP118 are named regions I and II, respectively. This section contains a set of aromatic and hydrophobic amino acids such as PHE 91, TRP 93, TRP 105, PRO110 as well as TRP134 of the β4 strand which participate in the ligand interactions [27]. Moreover, three other residues which are located at the regions I and II, ASP 84, GLU 87, GLU 113, and three amino acids that interact with Ca^2+^ such as GLU 109, ASN 111, and ASP 112, participate in interactions with ligand. Typically, calcium has eight coordination: five coordination with amino acids, and one bond with the water molecule. The presence of Ca^2+^ and the formation of suitable hydrogen bonds with amino acids provide the base for the binding of sugar molecules to this site. In the process of hydrogen bond formation, electron exchange takes place from an electron donor, which could be one sugar OH group, to the NH_2_ group of asparagine. Therefore, the presence of suitable amino acids as an electron acceptor is critical for hydrogen bond formation and binding of sugar ligands [28]. On the other hand, since the water molecule acts as a molecule with electron donor and acceptor capabilities, the presence of water molecules could facilitate the formation of hydrogen bonds between the lectin and sugar [29] (Table 2, Figure 4 and Figure 5).

### 2.7. Molecular Docking Analysis

Docking studies and comparison of the results of 1EGI and CTLM with L-fucose showed that the carbohydrate ligand of the CTLM binds to its receptor with high affinity and it is interesting that this interaction for the CTLM had a higher affinity in comparison to the reference molecule (Table 3). 

In the second docking, which was predicted between the WNV Envelop protein and the C-type lectin, the free energy value (Δ G) between the receptor and the ligand was compared. Interestingly, this evaluation confirmed the previous docking results as well (Table 4).

## 3. Discussion

*Culex pipiens* is a primary vector of WNV, maintaining local enzootic transmission among avian species [30,31]. This species exists under two biological forms, *pipiens* and *molestus* [32], and is abundant in temperate regions [33]. The females of the *Cx. pipiens* biologic form are capable of laying eggs without a blood meal in their first gonotrophic cycle, but those of the *Cx. pipiens* biologic form *pipiens* require a blood meal for reproduction [34]. In our study, all the collected *Cx. pipiens* belonged to *pipiens* form. We also could not find any *Cx. quinquefasciatus* species in the collected samples. This finding is accordant with the former investigations, which had reported the presence of *Cx. quinquefasciatus* in southern to central areas of the country [35].

It has been shown that many midgut proteins in mosquitoes are involved in interacting with arboviruses [36]. For example, multiple members of mosGCTLs act as the ligands to facilitate the invasion of arboviruses [13,14]. Regarding DENV, knocking down of the nine mosGCTL genes decreased the load of this virus in *Ae. aegypti* [13], suggesting that arboviruses may employ many mosGCTLs as ligands to infect mosquito vectors. In another investigation, the Japanese encephalitis virus (JEV) load was completely abolished when anti-mosGCTL-7 antibodies were injected [37]. Cheng et al., in 2010 have shown that the *mosGCTL*-1 and a protein tyrosine phosphatase (PTP) of *Ae. aegypti*), named *mosPTP-1* (*CD45* homolog, *AAEL013105*), act as a part of the same pathway which is important for infection of mosquitoes to the WNV. The mosPTP-1 participates in virus endocytosis, but its role as an attachment factor has remained unknown yet [38]. It has shown that mosPTP-1 is not critical for DENV infection [13]. Artificial feeding experiments showed that mosGCTL-1 antisera efficiently blocked WNV infection of *Ae. aegypti* and reduced its vector competence. It has been revealed that silencing of the *mosGCTL-1* does not influence DENV infection in *Ae. aegypti* which suggests that the product of this gene only facilitates WNV invasion [13,14].

In our study, *Cx. pipiens mosGCTL-1* was characterized using the primers which were designed with bioinformatics tools. The *mosGCTL-1* sequences were differed by 93% between the *Cx. pipiens* and *Cx. quinquefasciatus* (WGS: AAWU01017778). The intron length of the *mosGCTL-1* was varied and depends on the mosquito species: 61 bp in our study in comparison to 41 bp in the *Cx. quinquefasciatus* [14]. Furthermore, the open reading frame from the *Cx. pipiens mosGCTL-1* encodes 159 amino acid residues compared to the 165 in *Cx. quinquefasciatus* [14]. Regarding that the *mosGCTL-1* interacts with arboviruses, differences in the amino acid residues in different mosquitoes may affect the interactions of flaviviruses with this protein and consequently, these differences in the sequence may result in a structural difference and affect the vector competence of these species.

Our findings suggest that our produced polyclonal antibodies against mosGCTL-1 cannot efficiently block WNV compared to the previously performed study, which was carried out on *Ae. aegypti* that resulted in WNV infection blockage with the rmosGCTL-1 antibodies as well as the up-regulation of Cx. quinquefasciatus mosGCTL-1 by WNV infection [14].

Alignment and phylogenetic tree analysis of the *Culex pipiens* C-type lectins revealed that this protein has conserved residues that represent its key role in different species. The juxtaposition of different species of *Culex* and even the closeness of *Aedes* and *Drosophila* in the phylogenic tree confirms this idea (Figure 3). In this study, which is consistent with another study, which was conducted by Adelman and Myles in 2018 [39], it was shown that the conserved points not only observed in the extracellular part such as β4 sheet and region 1 but also are seen in the alpha-helix part inside the cell at the RANWKA sequence. Recognition of the conserved residues has particular importance in the optimal design of a small and complementary antigenic component. Effective activation of the central immune system is dependent on the presence of appropriate immunogenic components as well as the adequate antigen-Presenting cells. The presentation of processed antigens to the T-effector cells is mediated by the MHC molecules and the genetic diversity of the MHC plays a crucial role in the immune response [40]. Immuno-informatics analysis which was done using the NetMHCII showed that the peptide fragments of CTLM are relatively well presented to the immune system on the MHC molecule. It should be noted that in this analysis, lower values in Nano-molar affinity indicate a better presentation and create a more effective immune response. Our in silico results predicted that the HLA-DQA10102-DQB10602 is the best with the highest affinity among the common MHC molecules in Iran that could present the CTLM efficiently to the T-helper cells. It was also observed that some MHC molecules such as HLA-DPA10103-DPB10401, HLA-DQA10501-DQB10201, HLA-DRB1_0101, HLA-DRB1_0301, HLA-DRB1_0701, and HLA-DRB1_1501 are not efficient and some others are relatively efficient (Table 1). Therefore, it is not to expect a strong immune response in a wide range of the population. According to the predicted model and hydrophobicity analysis, CTLM has many conserved sites, which are located on the outer surface of the cell and are accessible to the immune system (Figure 4).

The superimposition of the CTLM molecule with the 1EGI as reference molecule showed that in general, these two molecules are very similar to each other in structure and the biggest difference between them is related to the Ca^2+^ interactive loop that its RMSD was ignorable and was about 0.426. On the other hand, the difference between a foreign antigen and host antigens should be such that it does not cause cross-reactivity with the native antigens and the development of autoimmunity [41]. Analysis and comparison of the final structure of the target protein (CTLM) and its human type (PDB accession number: 1TN3) showed that they have the least possible similarity. Their overall similarity is 29.85% and the similarity of the extracellular portion, which is exposed to the immune system, is about 19.26%. Therefore, it could not be expected that the injection of this antigen into humans may not be accompanied by the risk of inducing an autoimmune response. In this study, the protein-protein docking technique was used to determine the structural avidity of the CTLM as receptor and WNV as the ligand. The findings of this analysis predicted that the amount of free binding energy (Δ G) between the *Cx. Pepins* lectin (CTLM) and the WNV was significantly lower than the reference lectin (1EGI). In other words, the binding of *Cx. pipiens* lectin is much more effective to the virus (Table 4). This finding is consistent with the results of Focus-CTLM docking (Table 3). Other studies show that the virus binding to the receptor is pH-dependent [42]. Acidic environment (pH = 5) reduces the tendency of lectin to calcium and thus reduces the avidity of lectin–sugar binding. This process is closely related to the pattern of hydrogen bonding in the lectin protein surface (Figure 4 and Figure 5) [25]. Based on the findings of this study, it was predicted that the tendency of the virus to bind the lectin would be high. Moreover, due to the mosquitoes’ feeding, the acidic pH of the midgut would be reduced to close to 7.0, which causes the WNV to bind more effectively in competition with antibodies.

## 4. Materials and Methods

### 4.1. Mosquito Collection, Identification, and Rearing

According to the standard dipping technique (350 mL dipper), *Culicinae* mosquito larvae were collected using dipper and sometimes pipetted from Ghorogh forest (36°50′N 54°26′E) located in Golestan province, northeast of Iran. Mosquitoes were reared in the national insectary of Iran, malaria and vector research group (MVRG), and Pasteur Institute of Iran (Karaj, Iran). Larvae were identified using the key of Azari-Hamidian and Harbach [43]. Morphological characters as well as molecular identification were used to discriminate *Cx. pipiens* and *Cx. quinquefasciatus* species [35], and *Cx pipiens* species were reared under standardized conditions of the insectary (26 ± 2 °C, 70 ± 10% relative humidity, 12 h:12 h light:dark photoperiod). Larvae were placed in dechlorinated water supplemented with fish powdered food. Emerging adults were daily collected and transferred into cages and they were fed with a 10% glucose solution.

### 4.2. Characterization of the *Cx. pipiens mosGCTL-1*

The sequences of *Cx. quinquefasciatus* and *Ae. aegypti mosGCTL-1* (CPIJ010995 and AAEL000563, respectively) [14]) were used to design suitable primers for characterization of *Cx. pipiens mosGCTL-1*. As no match was found in BLAST Nucleotide collection (nr/nt) database (http://blast.ncbi.nlm.nih.gov/Blast.cgi), other databases of GenBank including Transcriptome Shotgun Assembly (TSA), Sequence Read Archive (SRA) and Whole-Genome Shotgun Contigs (WGS) databases were also used (WGS: AAWU01017778; SRAs: SRX1630418, SRX1630415, SRX1630413, SRX1630412, SRX1630411, SRX1630409, SRX1030166, SRX1030165, SRX1030164, SRX1030163, SRX1030162, SRX1030161, SRX1030160, SRX1030159, SRX870603, SRX870602, SRX870601, SRX806260, SRX806258, SRX806256, SRX804103, SRX626800, SRX626799, SRX565089; TSA: GFDL01000056). Primers were designed using the Gene Runner software (version 5.1.01d Betta, 1992-2015 Frank).

In the current investigation, total RNA and DNA of the reared *Cx. pipiens* were extracted from the midgut and salivary glands of female mosquitoes. Then, RT-PCR was carried out using the QIAGEN^®^ kits (Qiagen, Hilden, Germany). PCR reaction mixture contained 400 nM (each) primers, 1 unit of Taq DNA polymerase, 0.2 mM (each) dNTPs, 0.001% gelatin, 2.5 µL of reaction buffer, 1.5 mM MgCl_2_, and 1 µL of RT reaction product (for RT-PCR) in a total volume of 25 µL. Additionally, 100 ng of extracted DNA was used as the template to amplify the nucleotides containing the intron. A set of primers for *mosGCTL-1* amplification were designed: mos-F: 5′-CGACTTCGACTCAACATAAGCAG-3′ and mos-R: 5′-CTAAAACGGCGCCACACAATCG-3′. The optimized program was set as initial denaturation at 94 °C for 10 min, followed by 35 cycles of denaturation at 94 °C for 35 s, annealing at 58 °C for 35 s, and extension at 72 °C for 40 s, and an additional final extension at 72 °C for 10 min.

The PCR products were visualized on the agarose gel, and expected bands were recovered with a gel extraction kit (GeneAll^®^, Seoul, South Korea) and sequenced with an ABI 310 DNA sequencer machine (Macrogen^®^, Seoul, South Korea). Molecular and phylogenetic designing and analysis were carried out based on the bioinformatics tools such as Gene Runner (version 5.1.01d Beta), Chromas (version 2.6), and Mega7 (version 7.0) [44].

### 4.3. Cloning, Expression, and Purification of the *Cx. pipiens* Recombinant mosGCTL-1 (rmosGCTL-1) and Immunization of the Rabbits

Having the sequencing results, the following primers: mosmrnaF: 5′-GGATCCATATGTTGTCGAAAAGAAGTCTAGGAG-3′ and mosmrnaR: 5′-AAGCTTAAACGGCGCCACACAATC-3′ were designed for cloning process under the optimized amplification program: initial denaturation at 94 °C for 10 min, followed by 35 cycles of denaturation at 94 °C for 30 s, annealing at 58 °C for 30 s, and extension at 72 °C for 35 s, and an additional final extension at 72 °C for 10 min. The PCR product of *Cx. pipiens* mosGCTL-1 gene was cloned into the pET23a-plasmid using the *Bam*HI and *Hin*dIII restriction sites and expressed in *Escherichia coli* BL21 (DE3). The *E. coli* BL21-pET23a-*Cx. pipiens* mosGCTL-1-clone was grown in Terrific Broth (TB). When optical density (OD) at 600 nm was equal to 0.6–0.8, the expression was induced by adding the isopropyl-beta-d-1-thiogalactopyranoside (IPTG, Thermo Scientific, Waltham, MA, USA), and incubated for overnight.

The C-terminus His-tag fusion protein was purified with the Ni–NTA agarose (Qiagen, Hilden, Germany), and the eluates containing rmosGCTL-1 were desalted using Econo-Pac 10 DG columns (BioRad, Hercules, CA, USA). Bradford protein assay was used to measure the concentration of rmosGCTL-1 protein using a spectrophotometer (DeNovix, Wilmington, DE, USA) at OD_595nm_. The produced protein was analyzed after electrophoresis in 15% SDS–polyacrylamide gel (SDS-PAGE). Western blot assay was performed using the mouse anti-His antibody (Penta-His Antibody; Qiagen, Hilden, Germany) as the primary antibody, and Peroxidase conjugated anti-mouse IgG antibody (Qiagen, Germany) as the secondary antibody to confirm the purity of the rmosGCTL-1 (Figure 1). To produce polyclonal antibodies against the rmosGCTL-1, two female New Zealand rabbits (1–1.2 kg) were received from the laboratory animal science department of Pasteur Institute of Iran and housed in the animal care unit in standard cages with free access to food (standard laboratory rodent’s chow) and water. The rabbits were allowed to adapt for one week before the injection. The rabbits were immunized subcutaneously with the formulated rmosGCTL-1 (100 μg/rabbit/prime and 50 μg/rabbit/boost) in Freund’s adjuvant (Sigma-Aldrich Co., St. Louis, MO, USA), three times with 21-days intervals (complete Freund’s adjuvant was used for prime and incomplete Freund’s adjuvant was used for boost injection) [45]. For the control group, another rabbit with only the adjuvant and PBS was vaccinated. On day 63 after the first injection, blood was collected from both control and test rabbits, their sera were separated, and the production of specific IgG against the rmosGCTL-1 was analyzed and confirmed by Western blotting. The animal procedures were approved by the committee of animal ethics of Pasteur Institute of Iran (IR.PII.REC.1398.012).

### 4.4. Blocking Assay

Due to the unavailability of the BSL3 facility in Pasteur Institute of Iran and in collaboration with Institut Pasteur in Paris, after signing a biological material agreement; the egg rafts of *Cx. pipiens* were shipped under the optimal condition to the insectary of arboviruses and insect vectors unit, Institute Pasteur of Paris for WNV experimental infections as described before [46]. Seven-to-fourteen day-old of female *Cx. pipiens* were placed into the boxes, starved for 48 h in biosafety level 3 insectary (28 ± 1 °C, 80% relative humidity, 16h:8h light:dark photoperiod) and then were allowed to feed for 20 min through a pig intestine membrane covering the base of a capsule of the feeding system (Hemotek Ltd., Blackburn, UK) containing the blood–virus mixture which was maintained at 37 °C [47]. The infectious meal was composed of WNV, which was belonged to the lineage 1a and isolated from a horse in France (Camargue) in 2000 [48] suspension (1:3) diluted in PBS washed erythrocytes isolated from arterial blood of rabbit collected 24 h before the infection [46] and an equal volume of the serum removed from the washed rabbit erythrocytes was replaced by (i) control rabbit serum after immunization and (ii) immunized rabbit serum against rmosGCTL-1 for the transmission-blocking assay of control and test groups respectively (produced polyclonal antibodies titration assay showed affinity up to 1/102400). The ATP was added as a phagostimulant at a final concentration of 5 × 10^−3^ M. Virus titer in the blood-meal was at 10^7.3^ pfu/mL. After feeding, the fully engorged females were transferred into the cardboard containers and maintained with 10% sucrose at 28 ± 1 °C for 7 days. To evaluate the blocking efficiency of the anti-rmosGCTL-1 antibodies, the infection rate and Log RNA copies/body were assessed in the control and test groups at 0 (3 samples), 2 (20 samples), 5 (20 samples), and 7 (20 samples) days post-infection (dpi) by RT-qPCR method. Total RNA of the 126 infected mosquito whole body (head, thorax, and abdomen) were extracted using the Macherey-Nagel NucleoSpin^®^ RNA extraction kit (Hoerdt, France) to estimate the viral infection rate. Total WNV RNA was quantified by RT-qPCR using the Bio-Rad CFX96™ Real-Time PCR Detection System and Power SYBR^®^ Green RNA-to-C_T_ 1-Step kit. For each reaction, 10.55 µL of distilled water, 12.5 µL of buffer 2X, 0.375 µL (10mM) of each designed forward and reverse primers (WN175up: 5′-GTGTTGGCTCTCTTGGCGTT-3′ and WN259low: 5′-AGGTGTTTCATCGCTGTTTG-3′), 0.2 µL of mix-enzyme 125X, and 1 µL of RNA were used. The reverse transcription reaction was performed at 48 °C for 30 min. The qPCR conditions were 95 °C for 10 min, followed by 40 amplification cycles of 95 °C for 15 s and 60 °C for 1 min. Additionally, melt curve assay (65.0 °C to 92.0 °C: Increment of 0.5 °C 0:05) was carried out. For each run, the number of WNV RNA copies was calculated by absolute quantitation using a standard curve.

### 4.5. In Silico Studies

#### 4.5.1. Phylogenetic Prediction and Conserved Sequence Studies

First, by referring to the NCBI, the protein BLAST tool was used to search similar proteins to our query. Some subjects with higher similarity and scores, which were from different species, were selected to perform multiple alignments using the Clustal Omega [49] based on the Clustal W method to determine the conserved residues. In the next step, phylogenetic tree analysis was performed to show the racial relationship among the selected queries.

#### 4.5.2. Prediction of the Epitope Avidity

Proper presentation and effective stimulation of the immune system are the most important factors which should be considered for selecting and evaluating a vaccine candidate molecule. For this purpose, the antigen-presenting cells break down the antigenic epitopes into suitable fragments. Depend on the features of an antigen and presenting cells, antigenic peptides are broken into different pieces. These fragments are delivered to the effector immune cells by the major histocompatibility complex (MHC) system. Therefore, the NetMHCII 2.2 server was used to evaluate the potential properties of presenting the recombinant C-type lectin with the common MHC-II alleles in Iran. Iranian’s common MHC-II alleles were determined according to the previously performed studies [40,50]. The affinity value of the peptide fragments was determined using the NetMHCII. The predicted values are presenting in nanomolar IC50, and binding efficiencies are determined by the strong (Strong binder threshold 2.00) and weak (Weak binder threshold 10.00) indicators [51].

#### 4.5.3. Prediction of Protein Structure, Homology Modeling, and Molecular Interaction

Prediction of the 3D structure of mosGCTL-1 was performed using the SWISS-MODEL. The model design was accomplished based on the submitted primary amino acid sequence (GenBank: AUD11983.1). BLAST and HHBlits were used to find the most suitable patterns in the template library (SMTL version 2020-12-16) and maximum structure compatibility, respectively. Model construction was carried out using the ProMod3. Simulation of CTLM was done using the conserved residues and coordinating between the target sequence and the template. In the case of failed simulation by ProMod3, an alternative method, PROMOD-II, was used. Quality of estimated model and the final accuracy was evaluated using the QMEAN and GMQE, respectively [52,53,54,55,56].

#### 4.5.4. Molecular Docking Analysis

To evaluate an unknown molecule and its related interactions such as mosGCTL-1 of *Cx. pipiens*, one of the most appropriate and common technics is matching the target molecule with structurally characterized molecules. To achieve this goal, the C-type carbohydrate-recognition domain from the macrophage mannose receptor (PDB accession No. 1EGI) was used as the reference molecule. Predicted structure validation, superimposition, and molecular docking could be used for confirming the predicted structure. In molecular docking, target and reference molecules interact with a known molecule as ligand and the strength of the interactions are compared between the two molecules. In this study, to verify the predicted structure for the target molecule, protein–protein docking was used for the discovery of the receptor-ligand kinetics and responsible residues. According to the previously performed studies, l-fructose and d-mannose act as high-affinity ligands for the C-type lectin receptors [57]. Therefore, these molecules could be used as ligands for evaluating the lectin molecular interactions. In this study, l- fructose molecule (PDB accession No. 1K12) was used as a high affinity ligand. Protein-ligand docking was done using the Rosetta online server which is based on the scoring functions and search methodologies driven from the statistical analysis of the PDB. In this method, Metropolis Monte Carlo algorithm for identifying the binding sites and some other search strategies such as geometric hashing (FlexX), genetic algorithms (GOLD), systematic sampling (Glide), and physics-based force fields (Dock) are used for calculation and analysis [58,59].

Next, the probable interactions between the CTLM as a receptor and WNV envelope (PDB accession N. 2I69) as ligand were evaluated and simulated to find the most appropriate binding patterns with the lowest free energy. This analysis was accomplished using the pyDockWEB server that is the rigid-body docking software. In this method, free energy calculation is based on the Coulombic electrostatics with distance-dependent dielectric constant [60].

## 5. Conclusions

It could be concluded that our in silico investigation revealed that the WNV binds with high affinity to the *Culex* mosquito lectin receptor. The details of these interactions such as involved residues are given in Table 4. On the other hand, our blocking investigations suggest that other *Cx. pipiens* mosGCTLs presumably serve as receptors for the WNV and may act differently from its homolog in *Ae. aegypti*. Furthermore, additional investigations on the up-regulation of the *Cx. pipiens mosGCTLs* by WNV infection to develop an efficient TBV candidate are suggested. In this investigation, the rabbits were immunized with rmosGCTL-1, formulated in Freund’s adjuvant for the production of specific IgG. To improve the immunogenicity of subunit vaccines, using proper and potent adjuvant(s) could be a promising strategy to improve the pathogen-specific humoral, cellular, and functional immune responses [61].

## Figures and Tables

**Figure 1 pathogens-10-00218-f001:**
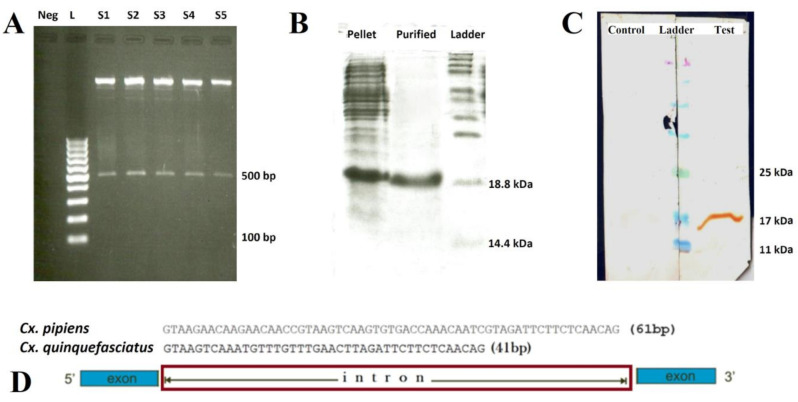
Cloning, SDS-PAGE, Western blotting, and intron sequence of *Cx. pipiens* mosGCTL-1; (**A**) Enzyme digestion (*Bam*HI, *Hin*dIII) of 5 colonies (S1–S5; S1 used in this study) of *E. coli* DH5α containing pET23a-*Cx. pipiens* mosGCTL-1-clone; L: 1kb ladder; (**B**) SDS-PAGE analysis of recombinant mosGCTL-1 that was expressed in *E. coli* BL21 (DE3) before (Pellet) and after purification (Purified) with Ni-NTA agarose; (**C**) Western blot was carried out using mouse anti-His antibody (Penta-His Antibody) as the primary, and Peroxidase conjugated anti-mouse IgG antibody (Qiagen, Germany) as the secondary antibodies to confirm the purity of the rmosGCTL-1 in control (vaccinated with only the Freund’s adjuvant and PBS) and test (immunized with the formulated rmosGCTL-1 in Freund’s adjuvant) rabbits. Sera from both groups were separated and the production of specific IgG against the rmosGCTL-1 was confirmed using this assay; (**D**) The intron sequencing result of *Cx. pipiens* mosGCTL-1: the length of *Cx. pipiens* mosGCTL-1 (MF361860, MF361861) intron (61 bp), compared to *Cx. quinquefasciatus* mosGCTL-1 (Cheng et al. 2010, CPIJ010995) intron (41 bp).

**Figure 2 pathogens-10-00218-f002:**
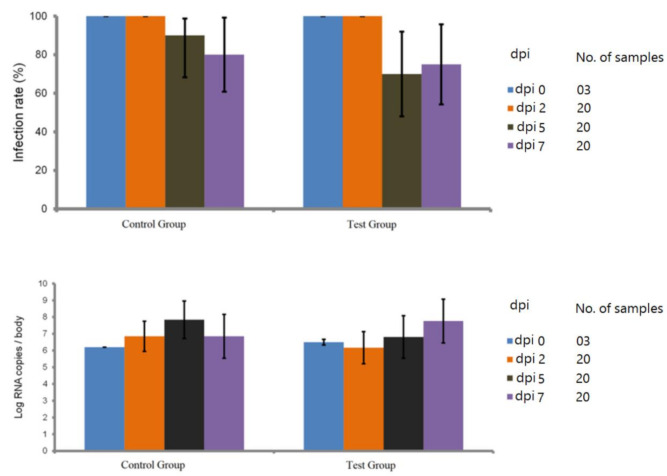
Low interference of rmosGCTL-1 antisera with West Nile virus (WNV) infection in *Cx. pipiens*. No significant differences in the infection prevalence as well as the Log RNA copies/body of studied *Cx. pipiens* among test group, compared with the control group were observed (*p*  >  0.05, Fisher’s exact test). For confirmation of the infection of mosquitoes to WNV at 0 dpi, 3 engorged mosquitoes were homogenized, and the presence of virus was confirmed, using RT-qPCR method. At 2 dpi, all of the mosquitoes in test and control groups remained infectious. At 5, and 7 dpi, the number of infected mosquitoes decreased in the test group (14/20), (15/20) compared to the control group (18/20), (16/20), respectively.

**Figure 3 pathogens-10-00218-f003:**
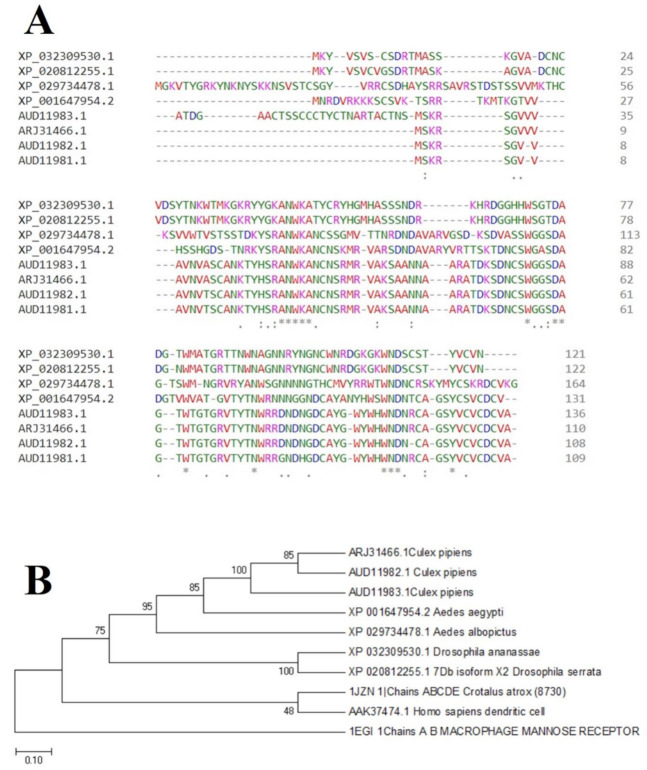
Alignment and phylogenetic tree analysis of the *Cx. pipiens* C-type lectins; (**A**): Alignment was carried out using the Clustal Omega based on Clustal W multiple sequence alignment method. Conserved residues were marked with an asterisk. One or two points indicate weak conservation; (**B):** Phylogenetic tree analysis of different organisms was performed using the MEGA7.0 software based on the neighbor-joining method with 1000 bootstrap replicates. The accession number of each subject has been shown in the front of the organism name. The repeat percentages of branches are shown next to each branch. The scale bar is equal to 0.1 changes per amino acids.

**Figure 4 pathogens-10-00218-f004:**
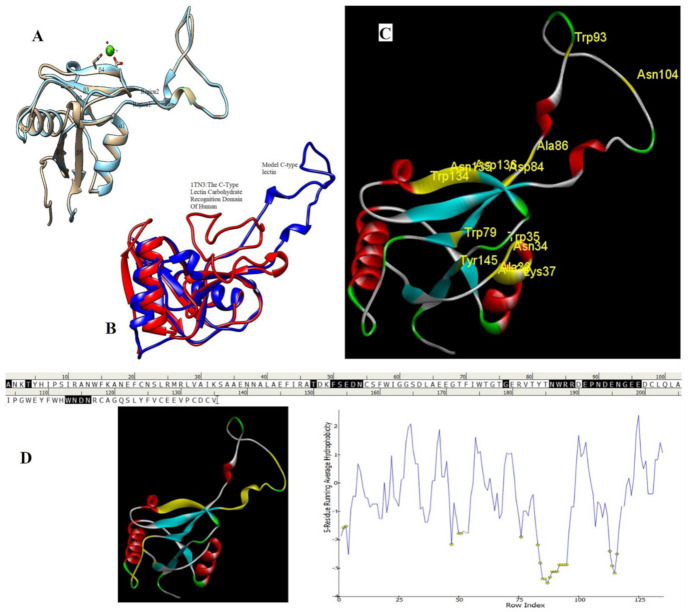
Superimposition of 1EGI and CTLM. The brown colored filament is 1EGI as a reference molecule, which its molecular properties has determined by crystallography. Light blue is CTLM and green is Ca^2+^ ion. By referring to the RMSD determination table, we can get that the greatest difference between the two strands is in the Ca^2+^ interactive loop. So that its RMSD was 0.426 and we can ignore such amount of difference (**A**); Superimposition of CTLM and PDB access No. 1TN3 as The C-Type Lectin Carbohydrate Recognition Domain of Human Tetranectin. This comparison which is very important in terms of host immune system arousal, show that the two molecules have little identity about 29.85 % in the sequence of amino acid (data not shown), in addition in terms of the structural availability to the immune system, also has a slight identity about 19.26% and the α- carbon matching of the two molecules indicates that their RMSD is 1.456 (**B**, **C**): According to the Clustal Omega alignment with some species of *Cx. pipiens*, *Ae. Aegypti,* and *Drosophila*, highly conserved residues are presented. Many of these residues are located in the cell surface so that exposed to ligands, but there is a conserved region in the alpha helix structure, which is located inside the cell and is out of reach of the ligand and the immune system; (**D**): Confirmation of hydrophilicity status and ligand accessibility. Residues have marked in black in the protein sequence row. The same residues have the lowest score in hydrophobicity, which is marked as yellow circles. The corresponding position of the residues, which interacts with the ligand, has marked as yellow in the schematic molecule. In fact, the low score of hydrophobic residues confirmed the availability of these amino acids to the ligand.

**Figure 5 pathogens-10-00218-f005:**
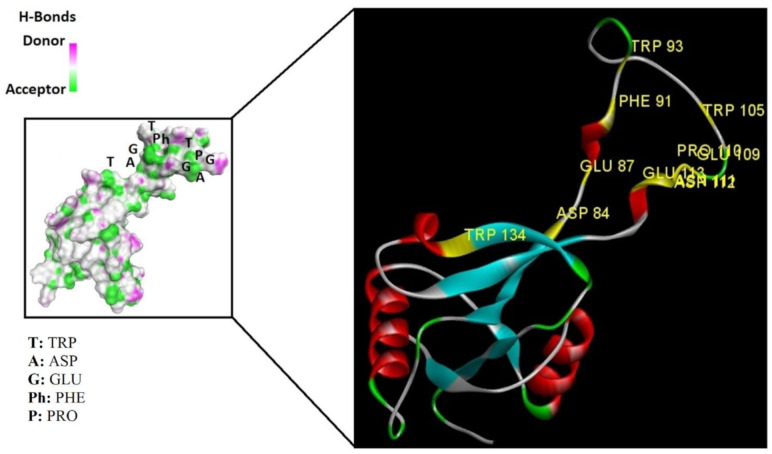
The interaction between the C-type lectin as a receptor and the ligand depends on the establishment of a hydrogen bond so in this interaction, some residues are donors and some are electron acceptors, which cause bonds formation. As shown, pattern of the green ring as electron acceptor, which are turning in the clockwise direction, confirmed the position of these amino acids in the molecular structure of the lectin.

**Table 1 pathogens-10-00218-t001:** Prediction of presented peptides of the C-type lectin model (CTLM) by common MHC-II alleles in Iran using NetMHCII.

No	MHC Alleles	Binding Situation	Highest Affinity (nM)	Number of Peptides
1	HLA-DPA10103-DPB10401	high binders = 0 weak binders = 0	-	122
2	HLA-DQA10101-DQB10501	high binders = 1 weak binders = 7	72.2	122
3	HLA-DQA10102-DQB10602	high binders = 7 weak binders = 11	11.9	122
4	HLA-DQA10501-DQB10201	high binders = 0 weak binders = 0	-	122
5	HLA-DQA10501-DQB10301	high binders = 4 weak binders = 22	29.0	122
6	HLA-DQA10501-DQB10303	high binders = 2 weak binders = 8	81.8	122
7	HLA-DQA10201-DQB10301	high binders = 1 weak binders = 42	62.3	122
8	HLA-DRB1_0101	high binders = 0 weak binders = 0	-	122
9	HLA-DRB1_0301	high binders = 0 weak binders = 0	-	122
10	HLA-DRB1_0401	high binders = 0 weak binders = 2	151.8	122
11	HLA-DRB1_0402	high binders = 0 weak binders = 3	977.3	122
12	HLA-DRB1_0403	high binders = 0 weak binders = 16	750.8	122
13	HLA-DRB1_0404	high binders = 2 weak binders = 9	26.5	122
14	HLA-DRB1_0405	high binders = 0 weak binders = 3	118.6	122
15	HLA-DRB1_0701	high binders = 0 weak binders = 0	-	122
16	HLA-DRB1_1101	high binders = 0 weak binders = 1	90.9	122
17	HLA-DRB1_1501	high binders = 0 weak binders = 0	-	122

**Table 2 pathogens-10-00218-t002:** Comparison of Root-mean-square deviation (RMSD) of functionally important residues between 1EGI as a reference molecule and CTLM of *Cx. pipiens* strain GhF mosGCTL-1.

Sequence/ Amino Acid	Cys735 Cys 119	Cys 759 Cys 148	PHE 708 to ASN 728 PHE 91 to ASP 112	11 Different Residues
Sequence importance	Disulfide bridge Core stabilizer	Disulfide bridge Core stabilizer	calcium interactive loop	Ligand interactive residues
RMSD	0.041	0.135	0.426	0.087
Figure	n	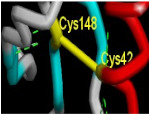	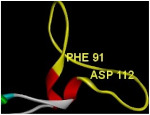	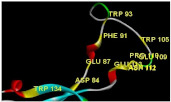

**Table 3 pathogens-10-00218-t003:** Docking analysis of l-fucose as high affinity ligand to the C-type lectin with 1EGI and CTLM as its receptor.

Molecule	Coordinated Amino Acids 2D Pattern	Δ G of InterfaceKcal/mol	Total Δ G Score (Kcal/mol)	Graph Pattern of All Interaction and Highlighted Lowest Score	Graphical Interaction View
**1EGI**	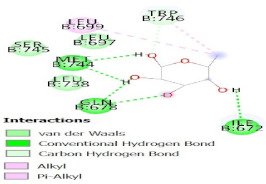	−5.4	−115.9	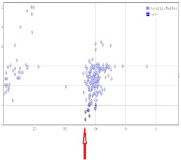	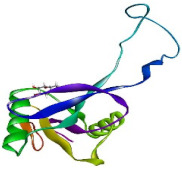
**CTLM**	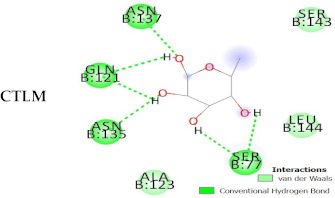	−3.9	−137.1	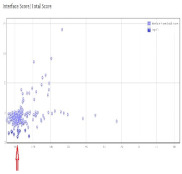	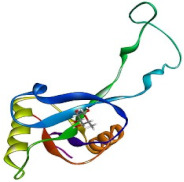

**Table 4 pathogens-10-00218-t004:** Docking analysis of 1EGI, CTLM, and West Nile virus (WNV).

Rank	1EGI 2I69 Involved Residues	Total Δ G (Kcal/mol)	Graphical Interactions View
**1**	SER711:THR318 ASP712:GLY319 TYR718:SER07 ALA722:ASP28 TYR723:ASP28	−98.762	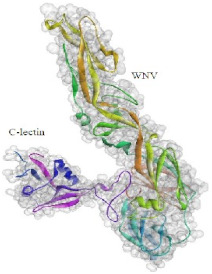
**2**	THR700:SER15 6TYR723:LGY334 GLY724:ALA365 GLY724:THR366	−93.892	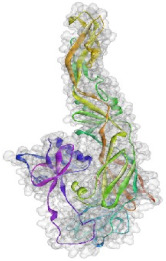
**3**	VAL732:SER156 TYR723:ARG166 SER705:ASN47 ARG663:GLY5 ALA664:GLU150	−91.021	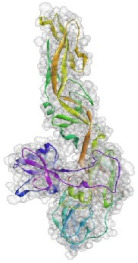
**Rank**	CTLM: 2I69 Involved residues	Total Δ G( Kcal/mol)	Graphical interactions view
**1**	GLY141:SER156 GLU113:GLY334 Glu113:ALA367 GLU390:GLY95	−102.560	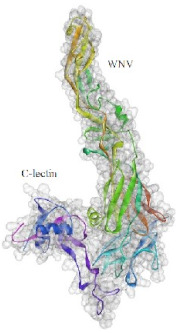
**2**	GLU129:ASP317 GLU129:GLU150 PHE131:GLY5 GLU98:SER276	−102.073	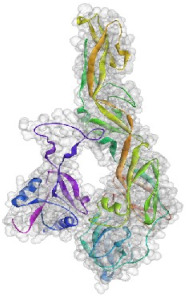
**3**	ARG107:ALA365 TYR102:GLU150 GLU88:THR366	−98.679	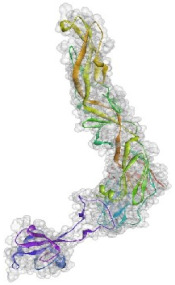

## Data Availability

Not applicable.

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
