# Peer review of "Developing a Vaccine to Block West Nile Virus Transmission: In Silico Studies, Molecular Characterization, Expression, and Blocking Activity of Culex pipiens mosGCTL-1"

_pathogens, 2021, doi:10.3390/pathogens10020218_

Round 1

Reviewer 1 Report

Bakhshi and others have submitted a manuscript entitled “Developing a vaccine to block…..activity of Culex pipiens mosGCTL-1”. The authors have shown a slight reduction in mosquito infection rate compared to controls. The docking results had shown a high affinity of WNV to lectin receptors. Also, the epitope avidity assay explained a high affinity of CTLM epitopes with MHC-II HLA-DQA10102-DQB10602. The study is well designed and had negative finding. However, this study has importance to understand the importance of lectin receptors in virus transmission.

The manuscript is well written, however, there are some issues that needs clarification.

Major comments:

  • In Fig 2, line 108-114: It is not clear why at dpi 0 the virus is present in the body. The explanation of dpi 0 is not well defined. In general, dpi 0 denotes before infection. Normally in in vitro culture the virus release starts from 8 hours post infection. Please clarify.
  • Table 1: It is not clear why MHCII (HLA-DQA10102-DQB10602) is important. No rationale or discussion has been included here. The authors should cite any relevant reference citing the importance of this allele and role in WNV pathogenesis will be beneficial for this paper.
  • A lot of redundancy in the discussion part (lines 239-241) with introduction (lines 68-71). Delete those sentences from introduction section.
  • Lines 237-238 can be regrouped with lines 64-68 and only discuss in the discussion section.
  • Lines 254-256 is redundant and to be deleted.
  • In blocking assay section (lines 394-427), antibody dilution and its characterization by in vitro neutralization assay needs to be shown, mentioned or properly referenced.

Minor comment:

  • Explain the PTP and CTLM when you use those words for the first time.

Author Response

On behalf of our co-authors, I would like to thank reviewer #1 for the critical reading of our manuscript. We made most changes suggested. Please, find below point-by-point answers to reviewer #1, indicating which changes have been made, and where they have been inserted in the text (highlighted in yellow).

Sincerely yours,

Hasan Bakhshi

Point #1: In Fig 2, line 108-114: It is not clear why at dpi 0 the virus is present in the body. The explanation of dpi 0 is not well defined. In general, dpi 0 denotes before infection. Normally in in vitro culture the virus release starts from 8 hours post infection. Please clarify.

We added and edited the following sentences (Lines 112-114): For confirmation of the infection of mosquitoes to WNV at 0 dpi, 3 engorged mosquitoes were homogenized, and the presence of virus was confirmed, using RT-qPCR method. At 2 dpi, all of the mosquitoes in test and control groups remained infectious.

Point #2: Table 1: It is not clear why MHCII (HLA-DQA10102-DQB10602) is important. No rationale or discussion has been included here. The authors should cite any relevant reference citing the importance of this allele and role in WNV pathogenesis will be beneficial for this paper.

We agree with the reviewer; the following sentences and reference were added to the text (Lines 134-138): In this algorithm, peptide fragments, which are more compatible with specific MHC II alleles, form stronger binds with less energy. A smaller amount IC50 (nM) indicates a better presentation of peptide to the T helper cells. Therefore, better presentation of antigenic peptides by antigen presenting cells causes better activation of the immune system against West Nile virus [51].

Point #3: A lot of redundancy in the discussion part (lines 239-241) with introduction (lines 68-71). Delete those sentences from introduction section.

We agree with the reviewer; redundant sentences (Lines 68-71) were deleted.

Point #4: Lines 237-238 can be regrouped with lines 64-68 and only discuss in the discussion section.

We regrouped lines 237-238 with lines 64-68 as follows (Lines 65-70): In 2010, Cheng et al., reported that WNV induces the expression of mosquito galactose-specific C-type lectin-1 (mosGCTL-1), and interactions between WNV and mosGCTL-1 are critical for infection of Aedes aegypti and Culex quinquefasciatus. In that study, generated antibodies against mosGCTL-1 of Ae. aegypti in rabbits inhibited WNV infection. Also, infection of Cx. quinquefasciatus (CPIJ010995) to WNV, resulted in up-regulation of mosGCTL-1 homolog in this mosquito [14].

We have discussed this study by more details in Discussion section (Lines 230-239).

Point #5: Lines 254-256 is redundant and to be deleted.

The redundant sentences (lines 254-256) were deleted; also, we edited the next sentences (Lines 257-261).

Point #6: In blocking assay section (lines 394-427), antibody dilution and its characterization by in vitro neutralization assay needs to be shown, mentioned or properly referenced.

Further to antibody dilution, we have mentioned the details in lines 402-410; regarding dilution information, we added the following sentence (Lines 409-410): (produced polyclonal antibodies titration assay showed affinity up to 1/102400).

Point #7: Explain the PTP and CTLM when you use those words for the first time.

It was done for PTP (Lines 237-238), CTLM (Line 130), and mosGCTL-1 (Lines 65-66).

Reviewer 2 Report

In this manuscript. Hasan et al., described a 30% blocking activity of The 30% blocking activity of Cx. pipiens mosGCTL-1 polyclonal antibodies (compared to the 10% in the control group) and a decrease in the infection rates of mosquitoes on day 5 post-infection, suggesting that there may be other proteins in the midgut of Cx. pipiens that could act as cooperative-receptors for WNV. Also, docking results revealed that WNV binds with high affinity to the Culex mosquito lectin receptors. This newly discovered vaccine may be effective in preventing WNV infection, and the purpose of this study is interesting. However, the results do not support the idea that mosGCTL-1 primarily interacts with WNV to promote viral infection.  Because in mosquitoes, mosGCTL belongs to a multi-gene family, and you did not analyze other mosGCTL paralogs that may act as susceptibility factors for WNV infection in mosquitoes. So we cannot support the mosGCTL-1 is the main factor promoting WNV infection in mosquitoes.

I have minor suggestions

  1. For abstract: “Additional investigations on the other mosGCTLs as transmission blocking vaccine can-33 didates against WNV are suggested”. I am not sure whether this sentence can be used as the “conclusion”.
  2. Figure 1C, the WB result should be improved.
  3. Figure 4-6 are only prediction and computation analyses, they can be merged as one figure.

Author Response

On behalf of our co-authors, I would like to thank reviewer #2 for the critical reading of our manuscript. We made most changes suggested. Please, find below point-by-point answers to reviewer #2, indicating which changes have been made, and where they have been inserted in the text (highlighted in yellow).

Sincerely yours,

Hasan Bakhshi

Point #1: The results do not support the idea that mosGCTL-1 primarily interacts with WNV to promote viral infection.  Because in mosquitoes, mosGCTL belongs to a multi-gene family, and you did not analyze other mosGCTL paralogs that may act as susceptibility factors for WNV infection in mosquitoes. So we cannot support the mosGCTL-1 is the main factor promoting WNV infection in mosquitoes.

We agree with the reviewer.

a) We added the suggested sentences in lines 33-35: Our results do not support the idea that mosGCTL-1 of Cx. pipiens primarily interacts with WNV to promote viral infection, suggesting that other mosGCTLs may act as primary infection factors in Cx. pipiens.

b) We did not analyze other mosGCTL paralogues because in 2010, Cheng et al., reported that WNV induced the expression of mosGCTL-1, and interactions between WNV and mosGCTL-1 are critical for infection of Aedes aegypti and Culex quinquefasciatus. In that study, generated antibodies against mosGCTL-1 of Ae. aegypti in rabbits inhibited WNV infection. Also, infection of Cx. quinquefasciatus (CPIJ010995) to WNV resulted in up-regulation of mosGCTL-1 homolog in this mosquito (Lines 65-70).

Point #2: For abstract: “Additional investigations on the other mosGCTLs as transmission blocking vaccine candidates against WNV are suggested”. I am not sure whether this sentence can be used as the “conclusion”.

We edited the conclusions in abstract section (Lines 33-35).

Point #3: Figure 1C, the WB result should be improved.

We included additional information to the WB results of Fig 1C (Lines 97-101):

(C) Western blot was carried out using mouse anti-His antibody (Penta-His Antibody) as the primary, and Peroxidase conjugated anti-mouse IgG antibody (Qiagen, Germany) as the secondary antibodies to confirm the purity of the rmosGCTL-1 in control (vaccinated with only the Freund’s adjuvant and PBS) and test (immunized with the formulated rmosGCTL-1 in Freund’s adjuvant) rabbits. Sera from both groups were separated and the production of specific IgG against the rmosGCTL-1 was confirmed using this assay.

Point #4: Figure 4-6 are only prediction and computation analyses, they can be merged as one figure.

Thanks to reviewer #2 for this suggestion. Figures 4, 5, and 6 were merged as one figure (Figure 4; Line 186).

Round 2

Reviewer 2 Report

The authors have changed/edited the manuscript according to my previous comments.

This manuscript is a resubmission of an earlier submission. The following is a list of the peer review reports and author responses from that submission.